# Impact of Microgravity and Other Spaceflight Factors on Retina of Vertebrates and Humans In Vivo and In Vitro

**DOI:** 10.3390/life13061263

**Published:** 2023-05-26

**Authors:** Eleonora N. Grigoryan

**Affiliations:** Koltzov Institute of Developmental Biology, Russian Academy of Sciences, 119334 Moscow, Russia; leonore@mail.ru; Tel.: +7-(499)-1350052

**Keywords:** vertebrates, human, eye, retina, spaceflight, microgravity, hyper-gravity, irradiation

## Abstract

Spaceflight (SF) increases the risk of developmental, regenerative, and physiological disorders in animals and humans. Astronauts, besides bone loss, muscle atrophy, and cardiovascular and immune system alterations, undergo ocular disorders affecting posterior eye tissues, including the retina. Few studies revealed abnormalities in the development and changes in the regeneration of eye tissues in lower vertebrates after SF and simulated microgravity. Under microgravity conditions, mammals show disturbances in the retinal vascular system and increased risk of oxidative stress that can lead to cell death in the retina. Animal studies provided evidence of gene expression changes associated with cellular stress, inflammation, and aberrant signaling pathways. Experiments using retinal cells in microgravity-modeling systems in vitro additionally indicated micro-g-induced changes at the molecular level. Here, we provide an overview of the literature and the authors’ own data to assess the predictive value of structural and functional alterations for developing countermeasures and mitigating the SF effects on the human retina. Further emphasis is given to the importance of animal studies on the retina and other eye tissues in vivo and retinal cells in vitro aboard spacecraft for understanding alterations in the vertebrate visual system in response to stress caused by gravity variations.

## 1. Introduction

Outer space is an environment alien to both humans and other vertebrates living on Earth. Such factors as microgravity (μg) and cosmic radiation make being in space dangerous. Long-term orbital flights often cause organic and systemic changes in the body, although they also trigger mechanisms of physiological adaptation to the extraordinary conditions of spaceflights (SF). Effects of μg on the redistribution of fluids, including blood, in the body are known. The cardiovascular system counteracts gravity when it pumps blood to the upper body, but it also uses the pull of gravity when it distributes fluid to the lower extremities. The conditions of μg alter the functions of the cardiovascular system and eye blood supply [1,2,3]. SF is known to have a negative effect on the central nervous system [4,5,6,7]. Moreover, muscle and bone mass loss, anemia, and immune system suppression were also documented [8,9,10].

Despite the protection created in spacecrafts, cosmic radiation, as a co-factor of SF, and μg exert a combined negative effect [11]. The eye and the retina, being one of the most sensitive systems of perception of the environment, are exposed to both [12]. Astronauts (cosmonauts, or taikonauts) during SF suffer a reduction and changes in vision, which are collectively referred to as spaceflight-associated neuro-ocular syndrome (SANS) [13,14].

In recent decades, numerous μg-associated pathological and physiological changes have been investigated at different levels of vertebrate organization, from single cells to a whole adult organism. However, only a small portion of studies have considered the effect of altered gravity on the visual system, the development and regeneration of eye tissues, and, in particular, the retina of animals in vivo. Beyond the sufficiently close investigation and elucidation of SANS in the literature, when analyzing available information, we find only scattered data obtained on different models of vertebrates in vivo, from fish to mammals. There is only scarce evidence about the behavior of retinal cells and the expression of genes and proteins in them when cultured in vitro under μg conditions. In many cases, these are results of laboratory in vivo and in vitro experiments simulating weightlessness close to physiological, the so-called simulated microgravity (s-μg). The range of issues to address using a vertebrate retina model in vivo under μg conditions is still also narrow. A few studies consider the retina development in fish and birds, the retina regeneration in amphibians, and changes in the retina of adult mammals, mainly small rodents. Data on the molecular mechanisms that provide the development patterns, regeneration/recovery of the retina, and its structural changes in SF conditions are even scarcer. There is also a very small range of studies that take into account the radiation effects in cases of μg exposure and on animals exposed to hyper-gravity (hg). Although the amount of information accumulated over more than half a century is not as sufficient, often scattered and fragmentary, it provides basic views of the effect that low gravity doses have on eye tissues: retina (both, neural portion of the retina and RPE), optic nerve (ON), choroid, lens, and cornea of human and other vertebrates. Currently, experiments using rodents are a priority, since the results of such approaches can potentially be transferred to a human eye model in order to prevent or address some vision problems faced by astronauts in SFs. However, all, without exception, studies using in vivo animal models conducted aboard a spacecraft are relevant for the development of fundamental aspects of the visual system biology in various animals exposed to SF conditions. Understanding the emergence and progress of changes, the potential and the pattern of adaptation, regeneration, and restoration of eye tissues, molecular mechanisms regulating these processes, etc. is of high importance. The already available information in these areas requires elucidation and generalization attempts to formulate the prospects and objectives of future studies. In this review, we attempted to briefly highlight the data obtained by studies on the retina of vertebrates from different taxonomic classes in vivo exposed to real microgravity (r-μg) on biosatellites and in manned flights as well as by on-ground experiments simulating μg and hg. Information about the changes occurring in the retina and surrounding tissues in humans in space missions is provided in brief. The review also contains information on the behavior of certain cell populations and retinal tissue in vitro in various cultivation modes simulating μg.

## 2. Factors Accompanying Spaceflights

The major factors that influence a living organism exposed to SF conditions are μg and ionizing cosmic radiation. Additional factors include hg experienced during takeoff and landing, magnetic field changes, an increase in CO_2_ concentration, circadian rhythm disturbance, vibration, behavioral and social constraints, etc., but the former two are the most influential (Figure 1).

Microgravity (μg) is a small dose of gravity at which the force of normal (1 g) gravity acts to only a low extent, and the body experiences weightlessness. SF conditions do not mean the complete lack of the gravity effect, but its dose is significantly reduced to values between 0.0001 and 0.000001× *g* (on average, 10^−6^) [15]. Over the years of research on board spacecrafts and after SF, as well as in experiments using s-μg, the adverse effect of long-term μg exposure on many systems of the body has become evident. Changes affect the visual, central nervous, musculoskeletal, cardiovascular, and immune systems as well as cell responses and the expression of many genes. Information collected on in vivo animal models, on humans, and isolated cells and tissues in vitro under s-μg and r-μg conditions has been summarized in numerous reviews elsewhere (see, for example, [9,16,17,18,19]). The data concerning the pattern of influence of r-μg and s-μg on the structures of the eye in general and the retina in particular are provided in the sections of the present review below.

Initially, it was assumed that cosmic radiation should have an effect on any living organism, including humans. This effect results from the loss of the shield created by the Earth’s atmosphere and magnetic field. In long-term SFs, the human or animal body is known to be continuously exposed to low doses of cosmic radiation including heavy ions (the so-called HZE particles) [20,21,22]. A study of the biological effect of space radiation on the mammalian body simulated in the laboratory has shown that it poses a higher cancer risk and negatively affects the functions of the cardiovascular, central nervous, and immune systems [23,24,25,26]. In vertebrates, including humans, the retina is very sensitive and subject to oxidative damage caused by the constant light radiation on Earth [27]. However, oxidative damage is an additional risk in SFs during exposure to cosmic radiation [28,29].

The variation in the effect of cosmic radiation in the conditions of μg exposure and the possible synergistic effect of the two factors are widely discussed [30,31,32,33]. Differences of the effect of space radiation on living organisms in SF compared to this effect at 1 g on Earth are documented from time to time, with information on this issue, however, being scarce, and the results still contradictory and ambiguous. This is explained by the greater attention to the role of μg and the fact that most studies are conducted without taking into account the cosmic radiation effect. Recently, researchers [11,33,34] have paid special attention to oxidative DNA damage and variations in signal transduction that occur during exposure to cosmic radiation, taking into account not only chromosomal DNA but also mitochondria. The study by Yatagai and Ishioka [31] proposes a solution for detecting the interactive effect of μg and space radiation using a broad analysis of gene expression. There is currently evidence that the combined effect of the two most important SF factors is exerted at the molecular level of cell responses: damaging and signaling by ROS, damage responses on DNA (repair, replication, transcription, etc.), and expression of gene and protein (regulation by chromatin, epigenetic control, etc.) [34]. However, it is obvious that this research approach requires modern methods of molecular biology to be introduced in the practice of experiments in SF conditions and/or when simulating the latter in a laboratory. The objective becomes even more complicated when using not only cell systems in vitro but also analyzing the molecular genetics of cell responses under the combined effect of radiation and μg on animal models in vivo.

The number of studies considering the effect of hyper-gravity (hg) on in vitro and in vivo models is not as substantial compared to that of studies on μg and cosmic ionizing radiation. On the other hand, astronauts experience, although for a short time, significant overloads during takeoff and re-entry. The effect of hg conditions is studied using short-arm human centrifugation as a possible countermeasure to treat not only spaceflight deconditioning but also provide a therapeutic approach to several pathologies [35]. During the experiment, many animal models have been studied, from insects to mammals including humans [36]. Nearly all body systems investigated are influenced by hg. Substantial anomalies have been observed in the cardiovascular, immune, vestibular, and musculoskeletal systems [37]. Thus, studies of the hg effect on changes in the mammalian musculoskeletal system show the occurrence of muscle hypertrophy, increased myogenesis, and inhibition of muscle degradation in vivo [38]. A similar effect on bone tissue has been recorded: elevation of bone mass due to the upregulation of bone formation [38]. The molecular mechanisms of the phenomenon have not yet been reported, but it is obvious that changes occur due to the regulation of genes responsible for these processes. In some of the studies, the hg effect on other tissues has been recorded: high doses of gravity can induce the synthesis of nitric oxide synthase (iNOS) in mouse kidneys [39] and disrupt the intestinal microbiota in mice [40]. A positive effect toward glial cells has been shown. Hyper-gravity promotes astrocyte reactivity aimed at suppressing axon dystrophy and stimulating neuronal regeneration [41]. Among numerous studies carried out to date using high gravity doses, no evidence of this effect on the visual system and eye tissues of vertebrates has been found except for the results of the below-described experiments using a centrifuge and newts as the animal model. Furthermore, there are large differences between the duration and values of hg created in laboratory-based experiments (days and weeks) and in SF (minutes). The body size should also be taken into account. As is assumed, the larger the body, the greater becomes its response to the impact [42]. For this reason, the data obtained in model experiments cannot be extrapolated to humans under conditions of overload during takeoff, subsequent SF, and re-entry to Earth.

## 3. Structure of the Vertebrate Eye and Retina

In the context of the review, we focus largely on the structure of the retina of the eye. The vertebrate retina is a multi-layer neuronal structure that converts light to electrical signals that are transmitted to the brain [43]. To perform these functions, the retina is highly organized [44,45]. In the evolutionary series of vertebrates, the retina has a common structure [44,45]. The laminated organization of the retina is schematically represented in Figure 2.

Outside, it is lined by a layer of retinal pigmented epithelial (RPE) cells that interacts with photoreceptors and provides simultaneously perception and processing of a light signal for transmission along the visual cascade and the optic nerve (ON) to the visual analyzer of the brain. The head of the ON is a region where the axons of the ganglion cell layer exit the eye. The visual cascade, in turn, is represented by cells of the nuclear (outer and inner, ONL and INL) and plexiform (outer and inner, OPL and IPL) layers as well as ganglion cells that extend their axons to ON. ONL is represented by bodies of photoreceptor cells, and INL is represented by bodies of interneurons, bipolar and amacrine cells, and also by horizontal cells. OPL contains processes of photoreceptors interacting with interneurons’ processes, and IPL consists of processes and synapses of interneurons interacting with ganglion cells’ processes. Müller glia (MG) is a population of macroglial cells within the retina. MG cells are distributed throughout the layer, providing mechanical support, sending their long processes outward and inward, and being involved in the formation of the outer and inner limiting membranes of the retina [46]. In addition, Müller glia cells provide trophic support for retinal neurons [47] and serve for the guidance of light through the NR [48]. In mammals and humans, the NR has two vascular supplies: the choroidal vasculature underlying the RPE and vessels of the inner retina (Figure 2). The blood supply to the inner retina is via the central retinal artery whose branches radiate from the ON head onto the inner retinal surface and then give rise to branches that extend into the retina through the INL, IPL, and OPL. Three interconnected plexuses in the inner retina and choroid provide oxygen and nutrients to neurons to maintain normal function of the retina [49,50].

## 4. In Vivo Experiments on Vertebrates during Space Flights and in Ground-Based Tests

### 4.1. Fish and Amphibians

Studies using animals from different taxonomic classes and species have always accompanied or preceded human space flights. Previously, the feasibility, expediency, and conditions of using animal models for research in long-term SF were analyzed [51,52,53]. Of particular importance are animals with short life cycles that can provide several generations while being aboard manned space stations. In this regard, fish offer ample opportunities to study the development and regeneration of tissues and organs, including the retina, due to the good knowledge, as well as the opportunity to carry out manipulations with molecular genetics on such model. However, very little is known about the retina of fish exposed to r-μg and s-μg conditions. The Japanese ice fish medaka was the object of investigation in a couple of studies [54,55,56]. The effect of μg on the medaka retina was studied at s-μg. The s-μg conditions are known to be different from r-μg, since s-μg platforms do not reduce gravity but constantly change its direction. Despite these fundamental differences, the s-μg conditions provide a generally accepted way to achieve physiological weightlessness in animals [57]. To study the organogenesis of the eye and retina, Nishiwaki et al. [54] used a 3D-clinostat and analyzed the general course of fish development including, in particular, the morphogenesis of developing retina. A comparison of the results obtained for the 3D-clinostat-treated group and the control group showed neither temporal differences in the retina development nor differences in the expression of the gene encoding opsin, which is a specific photoreceptor protein [54]. Subsequently, studies using mutant medaka (*Oryzias latipes*, Cab strain osteoblast transgenic fish) [58,59] were published. However, these publications did not provide information about the retina of fish exposed to both r-μg and s-μg conditions. A study based on this animal model aboard the International Space Station (ISS) was carried out later [60]. A wide range of issues concerning the development of organs and tissues of fish, including the brain and the eye’s retina, were addressed. Six-week-old male and female medaka were kept aboard the ISS in the Aquatic Habitat system for 2 months. Fish of the same line and age were also used for ground-based controls. Tissue morphology and the expression of a number of genes were studied by RNA-seq analysis. Histological studies did not reveal significant morphological changes in the development of tissues aboard the ISS. However, an RNA-seq analysis of 5345 genes for six different tissues indicated significant variations in the gene expression under r-μg conditions when compared to the respective ground-based controls. It was shown that the profiles of the stress-related GO genes change as the organism adapts to flight. In the brain and eyes, these turned out to be genes encoding 14-3-3 protein binding (GO:0071889), antigen processing and the presentation of exogenous peptide antigens via MHC class I molecules (GO:0002479), and also apoptosis-associated genes (GO:0006915). Thus, the major finding of the study [60] was the fact that the genes that change expression, including those in the retinal cells of fish, are the genes involved in adaptation to SF, which is associated with immune and stress responses of cells [60].

Studies at the cellular and molecular levels using modern methods and laboratory models of fish, including also various mutant fish lines, exposed to r-μg and s-μg conditions, are expected to be continued in the future. In this regard, we can mention the recent special design of a method and equipment for optical coherence tomography (OCT) to take images of changes that occur in the retina of s-μg-exposed adult zebrafish [61]. This device will allow analysis of changes in the eyes of not only fish but also other small animals, including animal models for the study of ophthalmological diseases.

To investigate the development in amphibians, Neff et al. [62] used tadpoles of *Xenopus laevis* and *Rana dybowskii* exposed to s-μg conditions at the hatching stage. An increase in the size of the animals’ heads and eyes was documented in the study. The authors associated such morphogenetic changes with variations in the cytoskeleton and/or proliferative activity of cells in the anterior region of the embryo. On the other hand, Savel’ev et al. [63] recorded a decrease in the sizes of the retina, ganglion of the VIII nerve, and olfactory placodes by 60, 22, and 17%, respectively, compared to the control group of developing *X. laevis*. The differences in the data obtained on ecaudate amphibian species may be explained by different factors: the developmental stages at which the animals were launched and the rearing devices used during the SFs.

Of certain importance is the suggestion by [64] that the effect of μg which causes changes in the expression of development-associated genes depends on the time point of development (the so-called “developmental windows”) at which exposed animals are particularly sensitive to μg exposure. The fact of significant changes in the expression of β-actin under s-μg conditions within 24–72 h post-fertilization is considered as an example. Another important suggestion is that μg during the amphibian development has a generally insignificant effect, and the resulting changes (differences compared to 1 g controls) are compensated after the removal of the impact, i.e., these are generally reversible [62]. Researchers [65] who studied the development of caudate amphibians exposed aboard the MIR station adhere to the same point of view. During all postflight development up to adulthood on Earth, no differences or abnormalities were found for the “flown” specimens returned to Earth.

Adult caudate amphibians are used to study regeneration in vertebrates in vivo. A group of researchers from IDB RAS has undertaken a wide series of experiments on newts (Urodela, family Salamandridae) (reviewed in [66]), tetrapods showing the highest regenerative potential [67]. Animals were exposed aboard the Russian Biosatellites and an unmanned “Foton” satellite. The participation in 12 flight-based studies jointly with IMBP RAS provided an opportunity to reproduce experiments and, thereby, obtain well-reproducible results [66]. Among the studied regenerating tissues, we used the lens and the retina. The data of studies in SFs and in experiments at s-μg, compared to the results of ground-based 1 g controls, allowed identifying specifics of regeneration of these tissues caused by μg. A study of lens regeneration showed an accelerated entry of iris cells (a source of lens regeneration) into the proliferative phase and also an increase in the proliferative activity of the cells at the stage of regeneration progress. This led to higher rates of development and growth of lens regenerates in the eyes of “flown” animals compared to the rates recorded from the ground-based synchronous controls. Furthermore, we found that the effects of r-μg and s-μg are long-lasting, rather mediated by body-wide factors and factors of the cell microenvironment than by direct ones [66]. Among the cell microenvironment factors, the FGF2 and HSPs signaling pathways were preliminarily identified [68]. In addition to lens regeneration, major focus was on the retina and RPE. The latter, located between the choroid coat and the neural retina, performs borderline functions, produces a wide range of regulatory factors, and is involved in the maintenance of visual function and absorption of the light energy, and the process is known as the visual cycle of the retina [69]. In addition, in adult amphibian and bird embryos, RPE cells are capable of changing phenotype to form neurons and retinal glial cells. After damage or removal of the retina, the whole retina is formed de novo through the proliferation and conversion of RPE cells [70,71,72,73]. In the r-μg conditions during a SF aboard Bion-11, we applied a model of cutting ON and blood vessels [74,75] (Figure 3A). With such damage, the retina initially undergoes cell death and degeneration. This activates RPE to be involved in the retinal regeneration. Simultaneously, the growth of undifferentiated cells occurs on the side of the growth region of the retina (*ora serrata*) in these animals. RPE cells enter the proliferation phase, undergo cell-type conversion, and form a population of multipotent cells subsequently differentiated into all types of retinal cells. In an experiment [74], we operated the newts at 2 and 4 weeks prior to an SF aboard the Bion-11 biosatellite, where the animals were then kept for 2 weeks. After the SF, the process of retinal regeneration was studied morphologically by radioautography (^3^H-TdR) and immunohistochemical methods. The most interesting results were obtained for animals operated 2 weeks before the SF. The proliferative activity of RPE cells in this group was 1.2–1.5-fold higher in the “flown” animals than in the synchronous 1 g control (Figure 3B). The *ora serrata* cells also showed increased proliferative activity compared to that in the 1 g control. The increase in the proliferation of source cells of retinal regeneration resulted in an accelerated retinal regeneration compared to ground-based controls [74,75]. Therefore, the study using the model for retinal regeneration after ON crosscut in Urodela animals exposed to r-μg has shown that the early steps of the process (RPE cell proliferation and reprogramming) are most sensitive to r-μg, which, in this case, has a positive effect on retinal regeneration. Thus, an assumption can be made that, both in the development and in the regeneration of the retina, there are periods of particular sensitivity to μg which are the stages of the progress of proliferation and the entry of retina anlage cells into differentiation.

Experiments were also set up using s-μg that was provided by the rotation of animals on a clinostat at 60 rpm. According to theoretical calculations, simulated microgravity could vary from 3.21 × 10^−2^ to 8.05 × 10^−3^ g. Sexually mature newts *Pl. waltl* were exposed to such conditions after microsurgical retinal detachment [76,77] (Figure 3C). It was found that the s-μg conditions do not prevent the NR regeneration after the detachment and, vice versa, have a positive effect on the regeneration. The animals with retinal detachment that underwent rotation on a clinostat, compared to the animals of on-desk 1 g controls, showed a decrease in cell death and an accelerated return of RPE to the original phenotype after the depigmentation, dedifferentiation and first steps of cell conversion that had occurred during the retinal detachment. In some cases, the s-μg conditions facilitated retinal reattachment, mostly in the dorsal and central part of the eye [76,77].

When discussing the identified r-μg and s-μg-induced features of regeneration/recovery of the newt retina, one can assume a generalizing effect exerted by the whole animal organism experiencing physiological weightlessness on the local regulators that control the regeneration processes [68]. These are, among others, the regulators identified in numerous ground-based in vivo experiments. Among the key ones, the FGF2, Wnt, Notch-Delta and other signaling pathways are known that work in cohort with the transcription factors of early eye development expressed and epigenetic regulators [78].

When studying the regeneration of eye tissues of newts exposed to r-μg and s-μg, we managed to obtain evidence of changes in the population of Müller glia (MG) cells of the retina [79]. MG, as a highly specialized cell population [80] showing certain resemblance with neural stem cells [81], is considered a potential resource for the vertebrate retina regeneration [82,83,84,85]. However, MG cells in mammals often respond to retinal damage by reactive gliosis. The latter is manifested as proliferation, an increase in the MG population, and cell hypertrophy [80]. Reactive gliosis accompanies many pathological conditions of the retina and serves to prevent glutamate neurotoxicity, in which MG cells produce trophic factors, including some that promote the survival of photoreceptor cell [86]. In experiments with retinal detachment under s-μg conditions, we revealed changes in MG associated with reactive gliosis: proliferation of its cells, an increase in the abundance of cell population, an increase in the thickness of accessory prolongations of MG cells, and increased expression of GFAP in them [79]. After 7 days (16 days p/o) of exposure of animals on a clinostat, the number of MG cells in the s-μg group was observed to increase 1.5–2.0-fold (with differences depending on the retinal region) (Figure 3D). The assessment was carried out on the basis of both the number of proliferating, [^3^H]-TdR labeled cells and the relative number of their accessory prolongations [77,79].

MG cell population is known to be influenced not only by growth factors and hormones but also by physical factors. It has been shown that the effect of laser-induced intraocular pressure (IOP) is accompanied by a significant increase in GFAP expression in the rat retinal MG population [87]. In humans who are in s-μg conditions [88], r-μg in SF [89], as well as under the influence of rapidly changing gravitational doze [90], variation in IOP was also observed. It is likely that in our case, the reactive state of MG in newts under the r-μg and s-μg conditions was also caused by IOP variations. Above are the results of an experiment aboard Bion-11 [74,79] using the model for ON cutting in newts. In this case, we also assessed the dynamics of variations in the number of MG cells on the basis of their long processes expressing GFAP in mature retinal regenerates in animals operated 4 weeks prior to SF. The number of MG cells turned out to be almost 40% greater in the “flown” newts than in the 1 g synchronous control. This again indicated the activation of the MG population in the de novo formed retinal regenerates. A model for lens regeneration after lensectomy, used in an experiment aboard the “Foton” M3 satellite, made it possible to study the MG state soon after a 12-day SF against the background of IOP decrease caused by the operation. Based on assessment of the intensity of GFAP^+^ immunospecific fluorescence in perikarya and processes of MG cells, a conclusion was drawn about the enhanced MG gliotic response in the SF group compared to the 1 g control. In the same localization in MG cells, we found the expression of HSP90, which is one of the cell stress indicators [91]. The intensity of expression of this protein in MG cells of the retina in the “flown” animals also turned out to be higher than in the 1 g control [79,92].

The available array of data provides ample evidence of the activation of MG cells in the retina of newts exposed to r-μg and s-μg conditions. The activation of MG toward the development of gliotic response is manifested, according to our experiments, as consistent cell responses: proliferation, self-reproduction, and also hypertrophy of cells with an increase in the thickness of GFAP^+^ processes. The results obtained indicate the development of reactive gliosis in the retina of μg-exposed newts. Thus, the extraordinary flight conditions and s-μg do not block, but, on the contrary, contribute to the regenerative processes in the Urodela retina after damage and pathological changes. This may be explained by the activation of the mechanisms responsible for maintaining and preserving the structure and function of the retina, including heat shock proteins [93].

In the USA–Russian joint study of the hg effect on the eye tissue regeneration in newts *Pl. waltl*, the focus was on the regeneration of the lens and also the tail and a limb [94]. However, the hg conditions unexpectedly exerted the most pronounced effect on the eye’s retina (Figure 4). Four groups of animals were used in the study. The first served as a “basal” 1 g control on day 9 post-lensectomy; the second was a control group kept in an aquarium; the third was subjected to centrifugation (hardware of Ames Research Center, 8 feet in diameter); and the fourth group served as synchronous (repeating all the conditions except hg) 1× *g* control. Centrifugation was conducted at 2× *g* (30.5 rpm) for 12 days. The right eyes of the animals of all groups served for non-operated control. No morphological features of the retina were revealed in the stationary on-desk 1 g control. Upon completion of centrifugation (day 21 p/op) in the 2× *g* group, the regeneration of the lens was significantly suppressed against the background of the retinal detachment recorded, which was presumably the cause of this suppression. The occurrence rate of disjunction of RPE cells and photoreceptors was about 70%. Retinal detachment was found not only in the operated eyes but also in the contralateral, non-operated ones. These cases in the 2 g group were evidence of the occurrence of detachment beyond association with lensectomy or other factors except hg. In cases of extensive detachment, there was retinal folding with cell death observed inside the folds. RPE cells initiated the conversion, migrated beyond their layer, and dedifferentiated. In some cases of NR detachment, the neovascularization took place on the dorsal side of the retina and in the ON head and bed regions. An antibody labeling against FGF2 and its receptors (FGF2R) in the 2 g group revealed a decrease in the expression of both the FGF2 ligand and FGFR2. In the centrifuged animals, the pattern of HSP70 expression also changed compared to the 1× *g* control. In the detached retina, the HSP70^+^ signal was additionally registered in INL and also in ganglion cells and ONs. It was assumed that the cases of retinal detachment at 2× *g* were caused by variations in the IOP level that occurred in centrifuged animals [94].

Under 2 g conditions, the healing of the cornea after its incision in the lensectomized eyes was inhibited: healing of the stromal and endothelial cornea layers was delayed. The inhibition of regeneration and recovery of eye tissues under 2 g conditions in newts is consistent with the information about the pattern of the hg effect on the amphibians’ development. Inhibition and abnormalities of development and growth were observed in experiments on the frogs *Rana rugosa* and *Xenopus laevis* that developed to the tadpole stage at hg (from 2 g to 10 g). Autopsy showed that brains, notochords, and muscles were reduced [95]. Patterns of the hg effect on the developing retina of ecaudate amphibians were identified in the study by Kawakami et al. [96]. The authors used the centrifugation of developing *X. laevis* embryos at 2× *g* and 5× *g*. They investigated cell death in the developing brain and eyes by the TUNEL method and gene expression using in situ hybridization. The 5 g group, compared to the 1 g control, showed a delay in the development in general and microcephaly and microphthalmia in particular. The developmental delay occurred against the background of high cell death and changes in the expression of the “developmental” genes *Xag1* and *Xag1*. However, no such significant changes at 2 g, as those at 5 g, were detected. According to the data obtained, hg can cause a slowdown in the overall development and also emergence of eye and retinal abnormalities, with the expression of the inhibitory effect of hg being dose-dependent [96].

Unfortunately, no other information about similar experiments and the effects of variations in the gravity dose on the retina regeneration processes in lower vertebrates has appeared in the literature to date. However, new data are desirable for two reasons. The first is the aquatic or semi-aquatic lifestyle of these animals (with neutral buoyancy) and, in this regard, their presumably different sensitivity to an altered level of gravitational load compared to terrestrial vertebrates and humans [97,98]. For this reason, understanding the differences in the effects of low and high g levels on the development of tissues, including the eye, would be a significant contribution to the animal gravitational biology research, in particular, to studies of adaptive and compensatory mechanisms. The second reason is that these classes of animals are indispensable objects for the regeneration research, since they exhibit the best regenerative potential and are well studied. The study of regeneration and mechanisms to control it under an altered influence of gravity is also one of the objectives of gravitational biology. Of particular importance is to identify the regeneration’s relationships with the age and size of the animal, the duration of flights, the gravity levels, space radiation, etc.

### 4.2. Birds

Birds have been used in experiments set up to study the embryonic development under r-μg conditions. For this purpose, the development of quail embryos is a convenient model by many criteria [99]. Fertilized eggs require only 17 days at 37 °C for development. A series of experiments with the early development of Japanese quail embryos (*Coturnix coturnix japonica*) exposed to r-μg conditions produced the following results: a slight delay in weight and body size gain [100], weak development of gastrointestinal tract tissues [101], and a delay in spinal cord development [102] and osteogenesis processes [103]. The disorders of eye morphogenesis were observed on the same model in an experiment aboard the Cosmos-1129 biosatellite [104]. Morphological and quantitative results were obtained for quail embryos aged 3 to 12 days (E3–E12). The results were compared to those for the groups of synchronous 1 g and laboratory-based controls. The main focus was on the study of the characteristics of anomalies in the eye development. Cases of abnormal formation of the inner layer of the cup, the prospective neural retina, were recorded at the optic cup stage, which is a key period in eye development (E3). The disorganization of neuroblasts and their migration toward the optic cup cavity was also observed here. On day 7 of development, during the period of active cell growth, significant morphogenetic changes were found in the eye anlagen of birds developing in SF. In some cases, small eye anlagen were formed that also had folding in the prospective neural retina and RPE. This indicated a dysregulation of retinal self-organization and its morphogenesis. Dysregulation was manifested as a reduction in the eye’s growth rate against the background of high proliferative activity of the anlage cells and also as a disproportional growth in the prospective RPE and neural retina. On day 10 of development (E10), as the eye tissues differentiated, the disorders detected on day 7 became even more pronounced. At that time, disturbances of the layered structure of the developing retina, a reduction in the vitreous body size, and a disturbance of the pecten oculi structure were recorded. The changes also affected the anterior part of the eye: the thickness of the cornea increased, and its layers became separated. It is worth noting that one similar case was found in the synchronous 1 g control. At stage E12, a microphthalmic eye that had hypertrophied choroidal and scleral coats was observed to develop in one individual from the SF group. The abnormalities of development of the quail eye, in particular the retina, recorded after SF were generally similar to the dominant types of eye development disorders in birds and mammals [105]. However, we still cannot state with certainty that these changes are directly related to r-μg. In this experiment, eye development anomalies were found in birds exposed to complex conditions aboard a biosatellite, where mechanical effects, among other ones, are also likely. Japanese quail embryos were again used to study the bird’s eye development in the USA-Russian joint experiment. Bird embryos were incubated at the MIR station at ages E14 and E16 [106]. Quantitative measurements, morphological, immunochemical analysis and electron microscopic studies were carried out to analyze the results of the experiment. Special attention was paid to the cornea: its dimensions, transparency, innervation, and ultrastructure were studied. As a result, no significant differences in the eye development were found in embryos developing under r-μg conditions compared to 1 g controls. The authors suggest that the lack of changes in the eye development in birds exposed to manned flight conditions is associated with the stable maintenance of IOP by animals during this developmental period and in SF conditions aboard the MIR station that differ from those aboard a biosatellite [106].

Previously, in a study of the hg effect, researchers used chicks at 2 weeks posthatch which were subjected to a 2× *g* hyper-gravity environment by chronic whole-body centrifugation for 7 days [107]. The animals were sacrificed at 3 weeks posthatch and then subjected to morphological and quantitative analyses. Compared to 1 g controls, 2 g exposure changed the parameters of retinal layers’ thickness: decreased mean widths of the ONL and INL, and also IPL were observed. However, the changes affected the ONL, OPL, and ganglion cell layers to a lesser extent. We could not find any other, more recent information about the hg effect on the condition of the bird retina, its development, or regeneration.

Despite numerous challenges encountered in setting up in vivo experiments on models of developing birds [108], it should be noted that the conditions of their life support and development were preliminarily well designed. For this reason, the use of bird embryos exposed to r-μg and/or to other doses of μg can be further expected. In these future studies, the range of issues of eye development’s relationship with changes in μg dose is also likely to be extended, for which bird models (quail, chick, and pigeon) can be very useful. Such studies in conditions of altered gravity could be as follows: the development and establishment of retinotectal projections, retinal regeneration in bird embryos, role of SF-related peroxidation and apoptosis of cells, behavior of MG, etc.

At the end of this section, it is also important to note that previously conducted studies on fish, amphibians and birds are within the framework of the fundamental questions being solved by gravitational biology. These studies are quite far from the direct, recent tasks of human stay in outer space. Direct extrapolation of the results to humans is impossible for many reasons—first of all because of evolutionarily fixed species features. However, the results seem important in a broad biological context.

### 4.3. Mammals

Significantly more studies have been conducted on mammals (mainly rodents) as models exposed to r-μg and s-μg conditions than on other vertebrates. This is largely because the results obtained contribute, to a greater extent, to the development of countermeasures for mitigating the SF effects on human eyes. Studies on mammals allow the prediction of changes causing optical complications in humans during SF, identify conditions for preadaptation, and, thereby, better prepare astronauts for long-term space missions. In early experiments by Philpott et al. [109,110] conducted aboard the Cosmos 782 and 936 biosatellites, first attempts were made to identify changes in the eye tissues of adult rats during a 3-week SF. The material was fixed shortly after the SF and on day 25 after the biosatellite landed. To test the effect of cosmic radiation, it was simulated by exposure to neon and argon radiation [110]. In general, the morphology of the rat retina fixed shortly after the landing did not differ significantly from that in the ground-based 1 *g* controls. However, necrotic cells were found in the ONL, and macrophage aggregations were present at the level of the OLM of the retina. On day 25 postflight, the state of the retina corresponded to the native one. Subsequently, the experiment on rats was modified: another control group of animals was introduced, which was subjected to centrifugation onboard to simulate 1× *g* conditions [110]. No differences were found between the flight-stationary and flight-centrifuged animals, but changes were seen between these two groups and the ground-based controls. The latter were similar to those recorded earlier [109] and those obtained during exposure to high-energy particles. Affected photoreceptor cells in the ONL showed swelling, clearing of cytoplasm, and disruption of membranes. As a result, a conclusion was drawn that μg and environmental conditions other than cosmic radiation do not contribute to the observed damage of retinal cells of rats [110]. In a study of eye specimens from rat pups “flown” in the NIHR1 and R2 spaceflight mission conducted by NASA, a histological examination of the retina showed no differences in development between the flight and control animal retina at E20, P1, P3, and P8 [111]. However, the retina of the experimental animals in the SF group was found to be much thinner [112].

After the experiments with developing neonatal rats exposed to s-μg aboard a space shuttle, the authors [113,114] concluded that there is a high probability of disturbances in the structure and function of the developing retina and, moreover, that the SF conditions aboard the shuttle can induce retinal degeneration in rats during development. On day 9 of SF, not only the retinas were at different stages of development in some of the individuals exposed for 3 days during their postnatal development, but also the outer segments of photoreceptors were not developed, which was accompanied by disturbances in the RPE underlying the photoreceptors. Significant disturbances were also recorded from GCL [113,114].

Recently, extensive research was conducted on mice exposed to r-μg conditions in space shuttle missions and under s-μg. An important fact was documented that cosmic radiation in combination with μg can induce oxidative damage in the rodent retina, leading to the apoptosis of retinal cells [28,29]. In an experiment set up by [29], mice were exposed aboard Atlantis (STS-135). The authors studied the expression of genes regulating the mitochondria-associated apoptotic pathway and also the levels of mRNA encoded by genes regulating the production of reactive oxygen species (ROS) in comparison with the data for 1 g on-Earth controls. As a result, an upregulation of ROS-associated genes was detected in tissue samples of “flown” animals compared to the 1 g controls. In parallel, the “flown” mice showed an increase in the number of apoptotic cells in the INL and GCL of the retina. As the authors suggested, μg in combination with cosmic radiation pose the major risk of retinal cell degeneration in astronauts after flights [29]. The integral effect of SF factors is also evidenced by other studies conducted also on rodents but with other tissues considered [115].

Thus, a study by Mao et al. [29] provided evidence that one of the important mechanisms of changes in the mouse retina during SF is a reaction similar to cellular oxidative stress. This assumption is consistent with the statements by Stein T.P. [116] and Yang et al. [117] that SF promotes peroxidation reactions in rodents and humans upon return to Earth under 1 g conditions. The effect is suggested to be more pronounced after a long-term SF and can persist up to a few weeks after spacecraft landing. In humans, an increased lipid peroxidation in erythrocyte membranes, a decrease in the blood level of antioxidants, and increased urinary excretion of molecules that are markers of oxidative damage to lipids and DNA were recorded. Observations on rodents showed an increased production of lipid peroxidation products and a decreased activity of antioxidant enzymes after SF [116,117].

Subsequently, Mao et al. [118] analyzed eye tissues to identify possible mechanisms of the r-μg effect on oxidative stress-associated retinal cell apoptosis as well as changes in the profile of expressed proteins in adult mice aged 9 months that were aboard the ISS for 35 days. The mice were kept under r-μg and, additionally, subjected to centrifugation on board (μg + 1 g), while a series of control 1 g experiments were set up on Earth. The data obtained for the groups of r-μg-exposed animals showed a 64% increase in the death of retinal vascular endothelial cells compared to the habitat 1 g control groups on Earth. According to the proteomic analysis, many key pathways responsible for cell death, cell repair, inflammation, and metabolic stress were significantly altered in the μg mice compared to the habitat 1 g control animals. A comparison with the μg + 1 g group animals also revealed differences in the expression of a number of regulatory proteins associated with the structure and metabolism of endothelial cells as well as with the immune response. The differences between the r-μg and μg + 1 g groups suggest that artificially created 1 g conditions aboard a spacecraft have a certain protective effect toward the vascular system of the mouse retina under SF conditions [118].

On Earth, the main methods simulating the redistribution of blood in the anterior part of the body that occurs under r-μg conditions are head-down tilt for human subjects [119] and tail suspension (TS) for small rodents [120] (Figure 5). In the study by Li et al. [121], prolonged TS was used for rats to reproduce the conditional μg effect on the redistribution of fluids in the cephalic region. After 35 days, the authors observed a number of significant changes in the eye and retina, in particular: the IOP changed, the choroid grew thicker, and the demyelination of ON occurred. At the cellular level, there was a decrease in the viability of retinal ganglion cells and ON oligodendrocytes. At the molecular level, inflammation-related factors were identified in the retina and ON. However, it is still unclear how the IOP changes during the experiment and whether IOP variations depend on the duration of TS of rats. The questions about the probability of compensation and reversibility of the detected disturbances also remain open [121]. These data agree well with the results of an earlier study [122] on the same model, TS, of rats. The changes occurring in the ON and retina were observed for 12 weeks. It was found that the conditions induce not only ultrastructural changes in the ON but also functional depression and substantial damage to retinal cells. In the ON, swelling of axons and disintegration of myelin were recorded. The number of apoptotic cells in GCL increased. The authors of both studies [121,122] regarded the findings as the result of changes in blood circulation in the cephalic region of rodents in general and the vascular system of the eyes in particular. Approximately the same issue was addressed in TS experiments on mice [123]. The state of the microvascular system was assessed on days 15 and 30 of experiment. After one month of experiments simulating cephalad shifting of blood, it was found that the exposure significantly changed microcirculation in the eye, in particular, in the ON head region, as well as electrophysiological parameters of the eyes. The authors emphasized that the changes recorded were only temporary and reversible. The mice could adapt to the changes in retinal microcirculation, which suggested the need to simulate conditions more similar to the outer space environment for retinal evaluation. The question was raised about the relationship of the severity of disturbances and probability of their reversibility with the time of s-μg exposure [123]. Theriot et al. [124] reported the results of the study evaluating the effects of hindlimb suspension (HS), one more analog of microgravity, on the rat retina. The study has been focused on the molecular and histological alterations in the retina. Several pathways and CSNK1A1-TP53 in particular were identified suggesting that stress is imposed by the HS treatment. It affected the retinal vasculature, oxidative and inflammation status, RPE function, and glial activation. The most significant genes showed gender- and age-specific expression for the first time. The IOP is regarded by the authors as one of the factors influencing the transcriptional responses in the retina [124].

Thus, changes in the state of rodent eyes against the background of cephalad shifting of blood can affect the microcirculation of the retina vascular system, provoke variations in IOP, and cause changes in the ON structure and the ganglion layer of the retina. The phenomenon of IOP variations in the altered dose of gravity is already mentioned above in the description of the studies on lower vertebrates, amphibians, and birds that were used in experiments with exposure to μg and hg conditions. It should also be noted that an earlier targeted investigation of this issue [125] revealed an increase in the IOP and in retinal vascular diameters of the vessels supplying the retina in 11 test subjects exposed to μg produced by parabolic flight onboard a KC-135 aircraft. The results showed that effects on the eye occurred very rapidly: within 20 s of exposure to artificial μg [125]. As noted in the literature, studies of conditions for IOP variations associated with μg in SF and in model experiments can help address the problem of ground-based ocular disorders such as glaucomatous optic neuropathy [126]. The phenomenon of oxidative stress and activation of ROS production with the subsequent apoptosis of retinal cells is another reason for the high risks posed by SF, μg conditions, and cosmic radiation. Studies of this phenomenon through experiments in SF and s-μg exposure are also a supplementary source of knowledge about the causes and consequences of oxidative stress of retinal cells and the occurrence of retinal degenerations associated with oxidative stress on Earth. The data obtained on rodents exposed to r-μg and s-μg conditions largely explain the changes observed in astronauts during SFs (see below). The effect of high g doses on the rodent retina has also been studied preliminarily. Kim et al. [127] considered the effect of 10 g on the IOP and retina of adult mice exposed to centrifugal acceleration for 4 h. After the centrifugation, IOP was measured, then the eyes were enucleated, and morphological and immunochemical studies of vascular endothelial growth factor-A (VEGF-A), VEGF receptor 1 and 2 (VEGF-R1,2), GFAP, and glutamine synthetase (GS) were carried out. The results indicated an increase in IOP with the lack of morphological differences in the retina of the centrifuged mice compared to the 1× *g* control. The levels of expression of the molecules studied were higher in the 10 g treated mice compared to the 1 g control. The authors conclude that the IOP, as well as the risks of hypoxic damage to the retina, increases at high g doses [127]. Table 1 summarizes the information obtained in long-term SF factors-related studies using vertebrate animals in vivo subjected.

### 4.4. Human

A vast array of information about changes in the human visual system during human orbital flights aboard space stations has been accumulated to date. In this section, we will only overview basic data, focusing in more detail on changes in the astronaut’s retina structure. In humans, ocular changes/disorders are considered as one of the major complications in a long-term SF aboard the ISS and when returning to Earth. Approximately 60% of astronauts who have been at the station for about six months and 29% of those in shuttles’ flights for two weeks manifest significant ocular changes that eventually lead to a decrease in visual acuity (see, e.g., [1,3,128,129,130]). After SF, a substantial percentage of astronauts have anatomical changes in the posterior eye tissues such as optic disc edema, globe flattening, and choroidal folds [131] (Figure 6). In particular, hyperopic drift with posterior flattening and choroidal folds are reported to directly affect the retina’s structure as a vision sensor. Changes in the vascular system feeding the retina [132] accompany IOP fluctuations [125]. The changes in ocular tissues in astronauts during SF, also known as visual impairment and intracranial pressure syndrome, affect their vision and ability to perform space operations. A combination of such changes is uniformly referred to as spaceflight-associated neuro-ocular syndrome (SANS) [3,130,133]. The observed changes are caused by the redistribution of fluid in the upper part of the human body and by the condensation of venous blood and lymph in the upper part of the body and the head. This phenomenon, in turn, is associated with the elimination of hydrostatic pressure gradients in the fluid-filled body systems under SF conditions [1].

The effect of altered hemodynamics in the eye’s retina was previously associated with biomechanical flows in choroid vessels in the posterior sector of the eye [134]. In the study of Nelson et al. [135], focus was on variations in IOP and intracranial pressure. When discussing the hemodynamic factor provoked by the conditions of gravitational changes, the authors noted the inconstancy of intracranial pressure, its potential to change depending on the position of the human body, the breathing pattern, etc. [136]. Thus, the major triggers of the observed ocular changes, recorded from astronauts during long-term SFs, are suggested to be orbital and cranial cephalad fluid shifts and SF-induced intracranial hypertension [1,137]. However, it is noted that there may actually be more physiological causes of the manifestation of SANS symptoms [130,138].

During long-duration SFs, nearly all astronauts exhibit changes within the spectrum of SANS. Their close relationship with structural changes in the retina has been specially analyzed [139,140,141]. Studies of [139,142] showed that the global total retinal thickness at the ON head, and also the peripapillary choroid thickness, significantly increase after long-duration SF compared to those before SF. In a study [143] based on the optical coherence tomography method, the peripapillary total thickness of the retina was evaluated as an early sign of ON disc edema in 19 crew members who had been aboard the ISS for 191 days. There was no strict association of changes in the retinal thickness with the development of optic disc edema in the astronauts after SF. The authors emphasized that other, optic disc edema-inducing factors should be considered in future research [143]. However, in a study by [140] using objective quantitative imaging modalities, the optical biometry of the structure of eye tissues was studied in 11 astronauts before, during, and after 6 months of SF. Parameters of changes such as peripapillary edema, axial length, anterior chamber depth, and refraction were taken into account. As a result, SF-associated peripapillary optic disc edema and choroid thickening in both eyes during early SF, which persisted throughout the mission, were recorded. It was traditionally noted that the altered ocular morphology, observed in association with SF, may be attributable to the chronic headward fluid shift that occurs immediately upon entering μg and remains throughout the duration of μg. Thus, despite some contradiction among the results obtained, the relationship of certain structural changes in the retina after cessation of the gravitational fluid pressure gradient and cephalad fluid shift in humans can be considered proven.

This key knowledge, as the result of ongoing research, helps to develop countermeasures for preventing or mitigating the risk of negative consequences for humans before and during long-term manned SFs [144]. Attempts have been made to understand how the effect of SF on the tissues of the posterior sector of the eye, ON head and bed, retina and its vascular system can be relieved [143,145]. Pardon et al. [145] used a short-term in-flight application of 25 mm Hg lower-body negative pressure and 10- to 20-min exposure. Lower-body negative pressure is a technique that redistributes blood from the upper body to the dependent regions of the pelvis and legs, thus reducing central venous pressure and venous return [146]. An acute exposure to 25-mm Hg lower-body negative pressure did not alter the ON head or retinal morphology, suggesting that longer durations of a fluid shift reversal may be needed to mitigate SF-induced changes. Marshall-Goebel et al. [143] tested three mechanical means of countermeasure separately and in various combinations to reduce the anterior shift of the fluid caused by the position of the body in a similar way as it occurs in SF. It was found that the countermeasures may need to be implemented for multiple hours a day. The essential issues that are currently discussed concern differences in the results of ground- and flight-based experiments and the necessity to fill these knowledge gaps for developing potentially reliable countermeasures in the future [130].

The findings as regards the cellular, tissue, physiological and functional features of the human eye’s state under r-μg and s-μg conditions are the vivid evidence of the dependence of the SANS manifestation on other systems of the human body, in particular the circulatory and lymphatic systems. Furthermore, the effects manifested during and after the cessation of μg exposure on a human presumably correlate and depend on the time spent in these conditions as well as on the individual characteristics of the body and its adaptability. Another challenge consists in the impossibility of clarifying the specific role of all factors and their combinations during SF (r-μg, hg and cosmic radiation, etc.) in the changes in the eye and, specifically, in the human retina recorded to date. The major challenge of identifying the causes and results of ocular changes also becomes obvious due to the lack of direct access to human eye tissues for research at the cellular, ultrastructural, biochemical, and molecular levels.

## 5. Studies of Retinal Cells In Vitro

In vitro studies on cultured vertebrate retinal cells, along with in vivo experiments, can contribute to understanding the effects of SF conditions on the vertebrate retina and the causes of SANS in astronauts. In vitro studies provide an opportunity to analyze changes in retinal cells at the molecular level, to disclose the molecular mechanisms mediating the effect of μg, radiation, and other factors during SFs. Exposure to s-μg in vitro is used when rotating tissues and cells in liquid media or on solid substrates for a variety of purposes and also in experiments with differentiated and stem cells of vertebrates [19,147]. Alterations in s-μg-exposed cells such as differentiation, adhesion, migration, and proliferation have been reported among other changes [17,18,148]. As mentioned above, RPE plays an essential role in maintaining the blood–retinal barrier, being involved in the metabolism of the visual cycle. There is evidence in the literature that s-μg can cause damage in human RPE cells in vitro, including changes in the cytoskeleton and gene expression [149,150]. A study by Roberts et al. [149] revealed pathological changes in isolated primary human RPE cells exposed to s-μg. S-μg (0.01 gravity) was provided by cultivating RPE cells in a rotating NASA-designed bioreactor for 24 h. The cells were analyzed at 48 h after rotation using comet assay and also by biochemical methods for detecting the level of production of prostaglandin E_2_ (PGE_2_), which is a marker of inflammatory response and a known risk factor for RPE cells. The results showed DNA breaks in RPE cells and the induction of PGE_2_ synthesis by them. Additionally, it was reported that the negative effect of s-μg on RPE cells can be mitigated/eliminated by pretreatment with cysteine, which is an agent having an anti-inflammatory effect. A human RPE cell line (ARPE19) was used in experiments in the Random Positioning Machine system simulating μg [151]. The results indicated negative effects of exposure on cells: decreased cell viability, apoptosis, blockage of the S-phase of the cell cycle, oxidative stress caused by increased ROS production, and activation of the Nrf2–HO-1 signaling pathway. These data, when considered together, become an example of the identification of molecular mechanisms acting in RPE cells under μg conditions and, with a high probability, also mediating the manifestation of SANS in humans in long-term SFs. A study by [152] was based on lines of human retinal cells and bovine aortic endothelial cells. These were co-cultivated under conditions of rotation in a bioreactor. As a result, a rapid (within 18–36 h) upregulation of vascular endothelial growth factor (VEGF) and FGF2 was revealed in retinal cells grown under s-μg as compared to monolayers. The authors [152] suggested that these experimental conditions contribute to accelerated capillary formation due to more effective three-dimensional cell assembly and differentiation. This, in turn, may be associated with the promoting of s-μg cell-to-cell interaction and/or secretion of growth and differentiation factors. Recently, Nguen et al. [153] have used the technique of clinorotation in a bioreactor simulating μg for cultivating a human RPE cell line ARPE19. A comparison of the results with the 1 g control in a static 2D culture has shown the formation of multicellular spheroids, a decrease in cell migration, and, additionally, an increase in intracellular ROS and mitochondrial dysfunction. Moreover, it has been found that s-μg activated autophagic pathways and also ciliogenesis. In a study of mitophagy activation, a possible trigger of the process is described: activation of the mTOR–ULK1–BNIP3 signaling pathway. The study has allowed suggesting a compound, TPP-Niacin, which is capable of effectively resisting the s-μg-induced oxidative stress and mitochondrial dysfunction. However, as is rightly noted in this regard, additional experiments are required on primary RPE cells in vitro and on animal models in vivo [153]. Son et al. [154] observed that s-μg stimulates the epithelial–mesenchymal transition (EMT) of human ARPE19 cells and induces vascular endothelial growth factor (VEGF) expression. The authors demonstrated also that an antioxidant Ishophloroglucin A could inhibit microgravity-stimulated or VEGF-induced EMT by reducing VEGF–VEGFR2 signaling [154].

A study using s-μg, generated by clinorotation, was carried out to identify changes in MG cells [155]. Observations were conducted during 15 min to 32 h of cell rotation in vitro. After a 30-min exposure, there was a disorganization of F-actin microfilaments and intermediate filaments of the cytoskeleton of MG cells. This, in turn, led to changes in the shape of MG cells, condensation of chromatin in them, and DNA fragmentation. It is worth noting that after 20 h of rotation in 1 g conditions, these negative changes were leveled out, the cytoskeleton was reorganized, and cells in the M phase were detected. In the study by [156], performed in vitro on primary MG cells of adult rats, natural antioxidants (aloin and ginkgolide A flavonoids) were tested in order to clarify their protective role in the cultivation of MG cells exposed to cosmic radiation. The latter was simulated using cosmic galactic rays at the Brookhaven NASA Space Radiation Laboratory. The results demonstrated a positive effect of antioxidants on the viability of MG cells through a decrease in ROS production in them. The authors propose this approach to mitigate the SANS manifestations and also to reduce the ROS production by cells of the aging retina of vertebrates and humans on Earth [156]. As mentioned above, endothelial vascular cells of the human retinal choroid are another retinal cell population (besides RPE and MG) highly sensitive to μg effects. Zhao et al. [157] cultured these cells for 3 days under conditions of rotation on a rotary system that provided μg. At the cellular level, the cells had a shrunk cell body, condensation and vacuolization of chromatin, mitochondrial cavitation, and apoptosis. It was also found that the s-μg effect on choroid vascular cells causes changes in the cellular ensemble, a decrease in the number of F-actin microfilaments, and activation of the Bcl-2 apoptosis pathway and the PI3K/AKT pathway [157]. Studies of retinal cells in vitro are summarized in Table 2.

We used an organotypic 3D rotational cultivation of whole-mount retinas. The retinas were isolated from the eyes of mature newts and adult Wistar rats. In the former case, cultivation was long term (for up to 30 days), and in the latter, it was for 10 days [158]. The major objective of the study was to clarify the behavior and role of potential cell sources of retinal regeneration (RPE and MG), while the rotation conditions were used only as a technique to improve the trophy of cells. Nevertheless, the data obtained can be considered, among other things, as a result of the effect of low-dose gravity compared to a static culture that usually demonstrates rapid cell death. Data on the retinal tissue culture showed an increase in the viability of retinal cells, an increase in the rate of tissue reconstruction in both animal species compared to in vivo conditions, and regenerative responses of cell sources of retinal regeneration/recovery [158,159]. In an analogous study, the formation of retinal organoids from mouse iPSCs was considered [160]. When setting up the experiment, the authors used the conditions of cultivation in a rotating-wall vessel (RWV) bioreactor, which creates not only optimal chemical environment for the development of the process but also low-g. As a result, retinal organoids were formed that reproduced the basic structure of the retina. It is important that the selected conditions had a stimulating effect on the rate of development of retinal organoids. The latter demonstrated a greater cell proliferation and a larger size compared to static cultures. The amplification of the cells continued as they differentiated, which led to an increase in the size of organoids by 40%. Although the authors [160] considered rotation conditions only as a way to improve the metabolism of cultured cells, such a consistency with our results seems at least noteworthy.

Special studies are also being conducted to determine the effect of ionizing radiation on the state of rodent and human photoreceptor cells under conditions of retinal explantation, cell cultivation, and γ-irradiation. These studies reveal species-specific differences in DNA repair in rod-photoreceptors following exposure to radiation [161], a greater degree of photoreceptor stability compared to other cellular types of the retina [162], as well as the dependence of the degree of photoreceptor damage on the chromatin structure [163]. These independent studies can make an additional contribution to understanding the combined effects of space flight factors, including ionizing radiation on retinal cells.

Despite the wide range of opportunities that cultivation with s-μg simulation in vitro provides, direct extrapolation of any data to an organism in vivo is not valid. As we can see from the data above, the results of the experiments vary depending on in vitro conditions, cultivation of single cells or whole-mount retinal tissue, cell types, animals from which cells were obtained, etc. It is also important that the s-μg effect on tissues and cells in whole organism in vivo is mediated by their systemic and local environment, which is significantly different from the culture medium. Nevertheless, there is no other way to study fine molecular mechanisms of the altered effect of gravity on cell populations that, in our case, are retinal cells. We should also note that, as in experiments in vivo, RPE, MG cells, and choroid vascular cells in vitro turn out to be populations of non-neuronal retinal cells highly sensitive to s-μg. These, along with GCL cells, fibers, and glial ON cells, will obviously be the focus of in vitro experiments.

## 6. Conclusions

The eye is the most important sensory organ for human and the overwhelming majority of vertebrate species. In SF conditions, during takeoff and adaptation of the animal and human body upon returning to Earth, the retina and the visual system in general are exposed to a combination of SF-related factors. The most important of these are variations in the gravity level and cosmic radiation. For more than half a century of human exploration of outer space, state of the eye has been among the most essential issues addressed by scientific studies in real SF and conditions simulating SF in ground-based experiments both in vivo and in vitro. Despite the relatively long history of the eye and retina research, today, it is possible to draw largely preliminary conclusions based on the results obtained. This is mostly due to the differences in goals and approaches, the wide range of animal models used and the low accessibility of SFs to set up experiments. In case of using animals developing in SF conditions (fish, ecaudate amphibians, and birds), despite some inconsistency of the data, there has been a probability of anomalies of the eye and retina development and, at the molecular level, changes in the expression of genes and proteins associated with cell stress and adaptation. The suggestions about the existence of certain stages of animal development at which the retina is most sensitive to the μg and radiation effects are important. Such a period in the vertebrate eye development is most likely the optic cup stage, which is a key one in the retina’s histogenesis. The issue of reversibility/compensation of emerging retinal anomalies during the further development of animals in SF and/or after returning to Earth remains unresolved. Studies on the retina of Urodela species have allowed a conclusion about the positive effect of SF and s-μg factors accelerating retinal regeneration, as well as the possible role of HSPs and FGF2 signaling pathways in this process. Furthermore, a probability of the gliotic response from the retinal MG cells was found. However, hg conditions provoke retinal detachment in these animals. Thus, variations in the gravity level and other SF-related factors can probably have a multidirectional effect, stimulating or inhibiting certain processes in the retina depending on the taxonomic class of animals, their age, size, conditions of μg exposure, and also the presumed initial pre-adaptation to μg existing in aquatic or semi-aquatic forms. 

In contrast to data on lower vertebrates, studies of changes in the mammalian (rodent) retina under r-μg and s-μg conditions yield more uniform results. They provide evidence of a high probability of oxidative stress, increased ROS production, and partial cell death in the retina caused by these processes. There is a general opinion that this phenomenon is caused by the cephalad shifting of blood occurring in SF conditions or simulated in TS tests. It is also the cause of changes in the vascular system of the retina and ON and is also manifested as inconstancy and variations in IOP. At the molecular level, an enhanced expression of stress- and inflammation-related genes is reported and discussed. In astronauts in SFs, the changes in ocular tissues, also known as visual impairment intracranial pressure syndrome, affect the vision and ability of space operations. A combination of such changes, referred to as spaceflight-associated neuro-ocular syndrome (SANS), is manifested, among other things, as some structural changes in the retina. Such manifestations as globe flattening, choroidal folds, choroid thickening, and peripapillary optic disc edema are known. 

The data obtained through in vitro experiments under s-μg conditions, when compared to stationary (1 g) cultures, provide evidence about the behavior of three important cell populations of the retina, RPE, MG, and choroid vascular cells, which are regulators of retinal functions and cell viability. The results generally confirm the findings in vivo such as the high probability of peroxidation, DNA damage, cell death, and proinflammatory response under conditions of μg exposure. However, studies of the retina development/regeneration processes or obtaining of retinal organoids in vitro showed that s-μg, which occurs in 3D rotational cultivation, vice versa, has a positive effect [164,165]. 

Further biological and medical studies of the retina in SF and under s-μg will focus on a few obvious aspects within the framework of space exploration and basic research. The former implies the retinal research aimed to predict negative effects of SF and the development of countermeasures for preventing or mitigating these effects in astronauts during long-term flights. The latter is an integral part of space exploration and makes a significant fundamental contribution to the development of gravitational biology, which is a vast field of science. In the future, it seems necessary (1) to study cellular and molecular events occurring in the vertebrate and human retina and in the visual system in general in response to effects of SF-related factors; (2) study the adaptive and compensatory mechanisms of retinal cells in animals and humans during SF and upon return to Earth; and (3) develop technical potential for conducting research by up-to-date methods, in particular by single-cell RNA sequencing and spatially resolved transcriptomics. The challenges and considerations relevant for robust experimental design to enable the growth of these methods in experiments under SF conditions are already discussed [166].

## Figures and Tables

**Figure 1 life-13-01263-f001:**
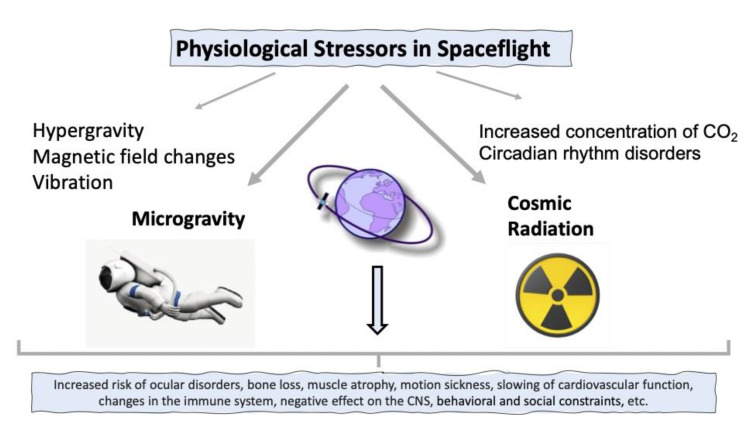
Spaceflight factors and their effects on vertebrates including humans. Microgravity and cosmic ionizing radiation are the major factors.

**Figure 2 life-13-01263-f002:**
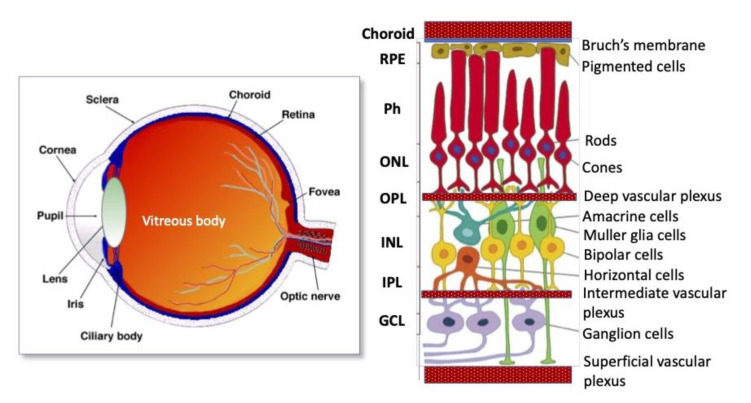
Diagram of the eye and retina: RPE—retinal pigmented epithelium; Ph—photoreceptors; ONL—outer nuclear layer; OPL—outer plexiform layer; INL—inner nuclear layer; IPL—inner plexiform layer; GCL—ganglion cell layer.

**Figure 3 life-13-01263-f003:**
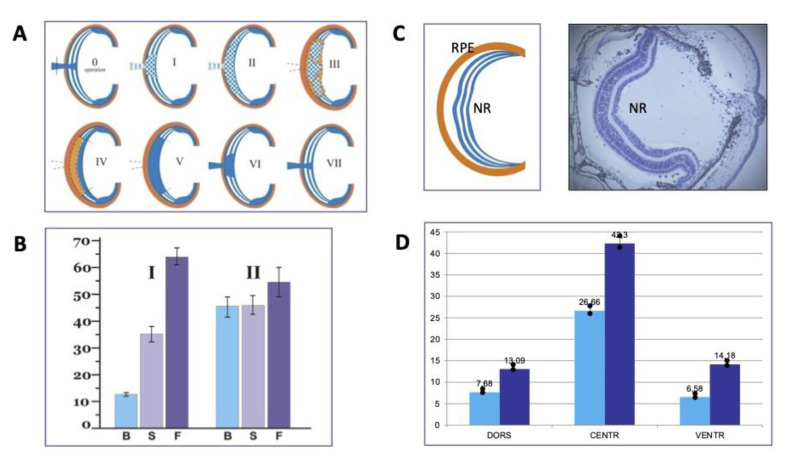
Spaceflight and simulated microgravity effects on Müller glia cell behavior in newts (Urodela). (**A**) Diagram of retinal regeneration after optic nerve crosscutting in newt: I, II—degeneration of the original retina; III, IV—RPE transdifferentiation and formation of the regenerating retina rudiment; V–VII—differentiation of newly formed retina and optic nerve regrowth. (**B**) Relative number (%) of cells expressing GFAP in regenerating retinas in newts of basal pre-flight control (B), synchronous on-desk control (S), and flight (F) in animal groups operated at 2 (I) and 4 weeks (II) prior to spaceflight. (**C**) A model of microsurgical retinal detachment used in clinorotation experiment. RPE—retinal pigmented epithelium; NR—detached neural retina. On the right, detached retina morphology. (**D**) Relative number of Müller glia cells’ long processes (accessory prolongations) in the dorsal (dors), central (centr), and ventral (vent) regions of the detached retina in clinorotated (dark blue) and control (light blue) animals on day 16 after retinal detachment and day 7 of rotation.

**Figure 4 life-13-01263-f004:**
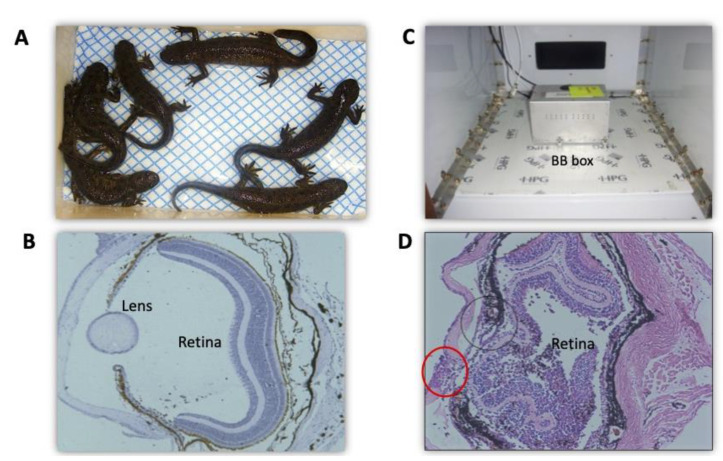
Hyper-gravity effect on the retina in lensectomized eyes of newts: (**A**) Newts (*Pleurodeles waltl*) used in the experiment; (**B**) BB box with animals placed inside the centrifuge chamber; (**C**) newt’s normal eye; (**D**) retinal detachment induced by 2× *g* centrifugation for 12 days; lens (outlined by black) and cornea (outlined by red) regeneration is inhibited.

**Figure 5 life-13-01263-f005:**
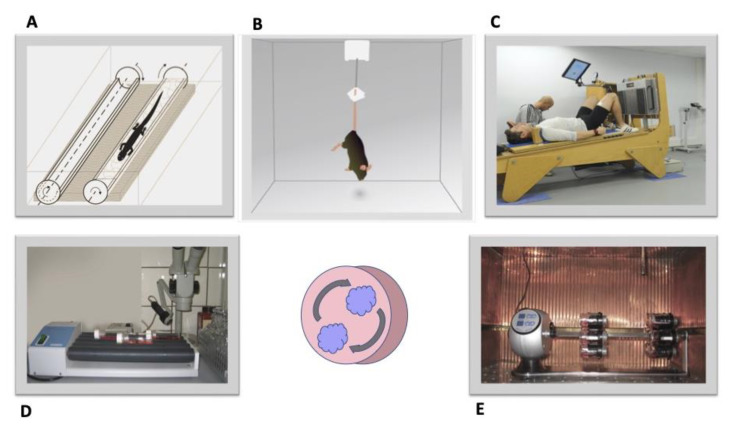
Various ground-based facilities for simulating microgravity in vivo and in vitro: (**A**) horizontally-oriented fast rotating clinostat that we used for caudate amphibians; (**B**) tail suspension (TS) model often used in experiments with small rodents; (**C**) head-down tilt used for human subjects to induce cephalad shifting of blood that takes place under microgravity conditions; (**D**,**E**) simple rotary systems for organotypic cultivation that we used in experiments on newt (**D**) and rat (**E**) whole-mount retinas.

**Figure 6 life-13-01263-f006:**
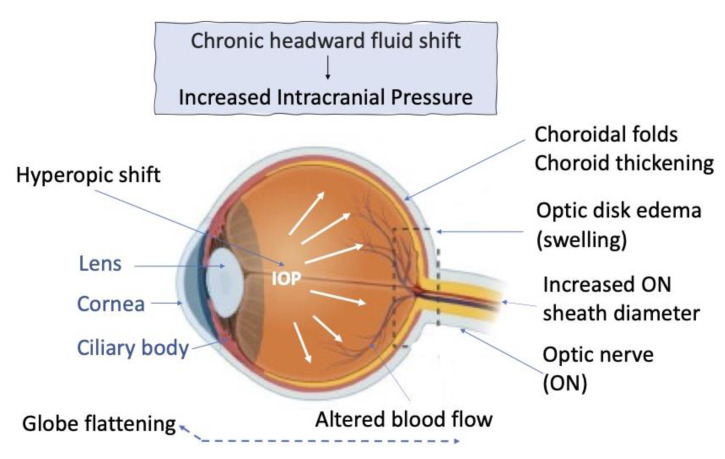
An overview of spaceflight-associated optic disorders found in astronauts’ eyes.

**Table 1 life-13-01263-t001:** In vivo experiments on vertebrate animals during space flights and in ground-based tests.

Animal (Class, Species)	Experimental Conditions	Research Direction
**Fish**		
Japanese Medaka*Oryzias latipes*	SpaceflightSimulated microgravity	Medaka model development [55,56,59]Gene expression [54,60]Eye, retina development [54]
**Amphibia**		
Frogs*Xenopus laevis**Rana dybowskii* 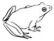	Simulated microgravity	Eye, retina development [62,63]
Newts *(Pleurodeles waltli)*	SpaceflightSimulated microgravitySimulated hyper-gravity	Retinal regeneration [74,75]Retinal detachment [76,77,94]Muller glia cell reactivity [79]
**Birds**		
Japanese quail *(Coturnix coturnix japonica)* 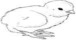	SpaceflightSimulated gravity onboard	Eye, retina development [104]Cornea development [106]
Chick (*Gallus gallus*)	Hyper-gravity	Eye, retina development [107]
**Mammals**		
RatMouse	SpaceflightSimulated gravity onboardSimulated microgravity	Eye, retina condition in adult animals [109,110,111]Eye, retina development [112,113,114]Retinal tissue and cells, molecular investigations [29,116,117,118]; Blood circulation [121,122,123,124,125]Eye, retina condition in adult animals,molecular investigations [127]
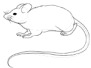	Hyper-gravity

**Table 2 life-13-01263-t002:** Studies of retinal and vascular cells subjected to microgravity in vitro.

Type of Cells	Kind of Experiment	Aims of the Study
Primary cell culture of human RPE	S-μg, rotating bioreactor	Cell death, inflammatory response, gene expression, cell protection [149,150]
Human RPE cell line (ARPE19)	S-μg, random positioning machine system	Cell viability, cell cycle, oxidative stress, ROS production, signaling [151]
-“-	S-μg, rotating bioreactor	Spheroid formation, cell migration, ROS production, autophagy, signaling, cell protection [153]
-“-	S-μg, rotating clinostat	Epithelial–mesenchymal transition, cell protection [154]
Muller glia cell line (C6)	S-μg, rotating on clinostat	Cytoskeleton changes, mitotic activity [155]
Rat Muller glia primary cell culture	Simulated cosmic Radiation	ROS production, cell viability, protection by antioxidants [156]
Human retinal cells and bovine aortic endothelial cells Vascular cells of the human retinal choroid	S-μg, cell co-cultivation, rotating bioreactor S-μg, rotating bioreactor	Growth factor expression, capillary formation [152] Apoptosis, chromatin condensation, signaling [157]

## Data Availability

Not applicable.

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
