# Peer review of "Impact of Microgravity and Other Spaceflight Factors on Retina of Vertebrates and Humans In Vivo and In Vitro"

_life, 2023, doi:10.3390/life13061263_

Round 1
Reviewer 1 Report
Review
The manuscript, “Impact of Microgravity and Other Spaceflight Factors on Retina of Vertebrates and Humans in vivo and in vitro” by Grigoryan et al. provide an overview of literature and authors’ own data to assess the predictive value of structural and functional alterations for developing countermeasures and mitigating the SF effects on the human retina. Further emphasis is given to the importance of animal studies on the retina and other eye tissues in vivo and retinal cells in vitro aboard spacecraft for understanding alterations in the vertebrate visual system in response to stress caused by gravity variations.
Comments and Suggestions for Authors:
1-Please correct the keywords according to MeSH.
2- Please summarize the contents of section "4. In vivo experiments on vertebrates during space flights and in ground-based tests" in a classified form in a table.
3- Lines 219-232 do not have references. Please refer to the reference of this study.
4- In lines 262-267, what is the meaning of "We used" at the end of line 262? Please mention the name of the author of the study. It is also said in line 267 "(see review: [67])". Please, if there is a need to mention information from other studies, reference should be given to that study while specifying that content.
5- As mentioned in comment 2, please summarize the contents of section "5. Studies of retinal cells in vitro" in a table.
6- The conclusion part of the article is very long. Please refer to the main results in this section.
Author Response
Reviewer 1.
First of all, I am very grateful to the reviewer for the attentive attitude to the review. Thank you for the specified shortcomings, recommendations for changes, additions, and corrections.
My answers and necessary explanations are given below.
Comments and Suggestions for Authors:
1-Please correct the keywords according to MeSH.
I’ve corrected the list of keywords in accordance with Mesh recommendation. However, I’m not sure, since the latter is not strong and depends on the journal
2- Please summarize the contents of section "4. In vivo experiments on vertebrates during space flights and in ground-based tests" in a classified form in a table.
Summarized information on Chapter 4 has been added to the text (marked) – Table 1.
3- Lines 219-232 do not have references. Please refer to the reference of this study.
The reference is added. (marked)
4- In lines 262-267, what is the meaning of "We used" at the end of line 262? Please mention the name of the author of the study. It is also said in line 267 "(see review: [67])". Please, if there is a need to mention information from other studies, reference should be given to that study while specifying that content.
The long-term experiments (1987-2008) involved researches of the Laboratory of Regeneration Problems (IDB RAS). We were given (by IMBP RAS) the opportunity to conduct a series of experiments aboard Russian biosatellites. The work was carried out on different models of regeneration (limb, tail, lens and retina of the eye). Also, our foreign colleagues took part in several studies. All references to the papers presenting the results of specific studies on the retina are given in the text of the chapter 4.1.
5- As mentioned in comment 2, please summarize the contents of section "5. Studies of retinal cells in vitro" in a table.
Table 2 “Studies of retinal cells in vitro" was added (marked)
6- The conclusion part of the article is very long. Please refer to the main results in this section.
The text of the conclusion has been shortened (marked in the margin)

Reviewer 2 Report
This manuscript reviews the impact of altered gravity conditions on the eye, in particular the retina. As manned space flight is expected to become more frequent in the near future, this is a relevant topic to address. Its quality could be improved, however, by making it somewhat more concise by avoiding too much repetition of content (such as lines 95-103), and by strictly focussing on gravity (the suggestion of ‘other spaceflight factors’ in the Title doesn’t seem warranted, even the factor cosmic radiation is hardly addressed and could, in my opinion, be removed from the manuscript). Also, I wonder if, in this type of review, a description of the eye and retina (chapter 3) is really necessary, and whether the issue of countermeasures should be discussed (after all, only one is mentioned in the entire manuscript). In the Conclusions section, it is said that effective countermeasures are being developed: which ones would that be?
Figure 1 (and accompanying text): I suppose that the behavioral and social constaints of (long-term) SF may be an important stressor as well?
The first sentence of the Conclusions (chapter 6) may be true enough when humans are concerned, but it cannot be extrapolated to vertebrates in general: e.g. infrared organs in snakes, echolocation in bats, tactile sense in moles, electroreceptors in sharks, olfaction in bears etc. etc. which may be much more important than vision.
When discussing the effect of s-µg, it should perhaps be stressed that gravity is never absent, its direction just ever changing.
Although, overall, the manuscript is fairly well written it should at some places be corrected or rephrased. A few examples below.
lines 17/18: I do not believe that structural and functional alterations have any predictive value in itself for developing countermeasures.
line 40: replace ‘reduce’ by ‘reduction’.
line 69: replace ‘exhibited’ by ‘conducted’.
lines 114-120: This section does not report anything substantial.
lines 121-122: replace ‘under the effect of’ by ‘during exposure to’.
lines 164-165: If this section is to be maintained (see above), this sentence should be rephrased.
line 183: In what way do Müller cells serve for light perception? See also line 281 for RPE cells: what is their function in light perception (except perhaps in relation to the circadian clock)?.
At various locations in the manuscript a phrasing such as ‘The object of research carried out by [54-56] was Japanese ice fish, . . .’ (line 205) is encountered. I would suggest to replace this by, for instance, ‘The Japanese ice fish was the object of investigation in a couple of studies [54-56].’ Please, check the manuscript for similar occurrences.
lines 355-356: is this about laser-induced ‘increased’ IOP?
line 381: replace ‘vice versa’ by ‘on the opposite’.
More than once the phrase ‘dose of gravity’ is used. I would suggest to use ‘magnitude of gravity’ or ‘gravity level’. I understand that one can study the dose-response of varying gravity but somehow ‘dose of gravity’ doesn’t sound appropriate to me.
line 529: should it be ‘naive’ instead of ‘native’?
line 759: What is meant by ‘metabolism of visual circulation’?
line 827: replace ‘analogical’ by ‘analogous’.
Author Response
Reviewer 2.
I express my deep gratitude to the reviewer for the work of reading and all the comments and suggestions made. Below – the list of questions and my answers for each of them.
Comments and Suggestions for Authors
This manuscript reviews the impact of altered gravity conditions on the eye, in particular the retina. As manned space flight is expected to become more frequent in the near future, this is a relevant topic to address. Its quality could be improved, however, by making it somewhat more concise by avoiding too much repetition of content (such as lines 95-103), and by strictly focussing on gravity (the suggestion of ‘other spaceflight factors’ in the Title doesn’t seem warranted, even the factor cosmic radiation is hardly addressed and could, in my opinion, be removed from the manuscript).
The sentences (lines 60-62; 95-103) were removed.
I could not follow the recommendation to remove other (besides microgravity) factors of SFs from the title and consideration. First of all, since SF effect assessment necessarily assumes their joint and combined contribution, despite the fact that separate effect each of them cannot be evaluated. Such evaluation is extremely difficult and, moreover, there is often a lack or inaccessibility of aggregate information about radiation levels, gravity level during takeoff and landing, etc.
I tried to mention all the results of SF-associated special studies on the role of cosmic radiation and hyper-gravity in the review context (Radiation: 119-134; 135-162;534-544;562-574;856-863. Hyper-gravity: 390-438; 500-506;664-673).
Also, I wonder if, in this type of review, a description of the eye and retina (chapter 3) is really necessary, and whether the issue of countermeasures should be discussed (after all, only one is mentioned in the entire manuscript).
I considered necessary a brief description of the vertebrate retina, since the responses of its individual compartments (including blood vessel system) and cell-types are the main subject of consideration in the paper.
If not discussed at all some countermeasures which are being developed, then it’s not clear what for experiments on mammals should be carried out.
In the Conclusions section, it is said that effective countermeasures are being developed: which ones would that be?
I’ve removed the sentence on countermeasures from the Conclusions (marked).
This issue is the subject of a special discussion/paper. In my paper I could just show some examples available in the literature and demonstrating that "countermeasures are being developed".
I can't offer my own recipes. Their formulation depends on many factors, for instance: conditions inside SS, the duration of SF, the individual characteristics of astronauts, technical capability and availability, specific suggestion of ophthalmologist, etc.
Figure 1 (and accompanying text): I suppose that the behavioral and social constraints of (long-term) SF may be an important stressor as well?
I included these factors in the list in Figure 1 (marked).
The first sentence of the Conclusions (chapter 6) may be true enough when humans are concerned, but it cannot be extrapolated to vertebrates in general: e.g. infrared organs in snakes, echolocation in bats, tactile sense in moles, electroreceptors in sharks, olfaction in bears etc. etc. which may be much more important than vision.
I’ve clarified: “The eye is the most important sensory organ for human and the overwhelming majority of vertebrate species” instead “The vertebrate and human eye’s retina is the most important sensory organ.
When discussing the effect of s-µg, it should perhaps be stressed that gravity is never absent, its direction just ever changing.
This main difference between s-µg and r-µg is noted (added, lines 208-211)
Comments on the Quality of English Language
Although, overall, the manuscript is fairly well written it should at some places be corrected or rephrased. A few examples below.
lines 17/18: I do not believe that structural and functional alterations have any predictive value in itself for developing countermeasures.
This phrase, when using the wording "... to assess the predictive value ..." in it, is a cautious assumption only, which does not mean that everything found in experiments can directly serve for countermeasures’ developing”.
line 40: replace ‘reduce’ by ‘reduction’.
Replaced and marked in the margin
line 69: replace ‘exhibited’ by ‘conducted’.
Replaced and marked
lines 114-120: This section does not report anything substantial.
This phrase is given in order to indicate, without going into details (the latter are in the links to the relevant papers), that this assumption is being discussed. At every and each conference I was able to participate, the section devoted to the synergistic effect of radiation and microgravity was necessarily organized. It is very important question. The difficulty lies in understanding the combined effect of these two factors. Thus, I thought it appropriate to mention.
lines 121-122: replace ‘under the effect of’ by ‘during exposure to’.
Replaced and marked
lines 164-165: If this section is to be maintained (see above), this sentence should be rephrased.
The sentence is rephrased: “The vertebrate retina is a multi-layer neuronal structure that converts light to electrical signals that are transmitted to the brain”
line 183: In what way do Müller cells serve for light perception?
A novel functional role of Müller cells, discovered in the last decade, is the guidance of light through the NR. There are some studies (doi: 10.1073/pnas.0611180104 doi: 10.1016/j.exer.2018.05.009; doi: 10.1038/ncomms5319) showed that light propagation by Müller cells through the retina can be considered as an integral part of the first step in the visual process, increasing photon absorption by cones while minimally affecting rod-mediated vision.
See also line 281 for RPE cells: what is their function in light perception (except perhaps in relation to the circadian clock)?
To be more precise: RPE closely interacts with photoreceptors in the maintenance of visual function and absorbs the light energy focused by the lens on the retina. It is the participant of the process is known as the visual cycle of retinal. The sentence rewritten (lines 279-282, marked)
At various locations in the manuscript a phrasing such as ‘The object of research carried out by [54-56] was Japanese ice fish, . . .’ (line 205) is encountered. I would suggest to replace this by, for instance, ‘The Japanese ice fish was the object of investigation in a couple of studies [54-56].’ Please, check the manuscript for similar occurrences.
Sentences were rewritten (marked)
lines 355-356: is this about laser-induced ‘increased’ IOP?
In the study [87] (DOI: 10.1167/iovs.02-0255) “Significant elevation of IOP from 1 to 7 days and loss of cells in the RGCL from 3 days onward were noted after trabecular laser photocoagulation”. So, it is about laser-induced ‘increased’ IOP?
line 381: replace ‘vice versa’ by ‘on the opposite’.
‘Vice versa’ replaced by ‘on the opposite’
More than once the phrase ‘dose of gravity’ is used. I would suggest to use ‘magnitude of gravity’ or ‘gravity level’. I understand that one can study the dose-response of varying gravity but somehow ‘dose of gravity’ doesn’t sound appropriate to me.
I replaced ‘dose of gravity’ by ‘gravity level’ throughout the text (marked)
line 529: should it be ‘naive’ instead of ‘native’?
Both words originally derived from the Latin ‘nativus’, an adjective meaning inborn or natural (from quora.com). So, I guess both variants are correct. ‘Native’ seems more frequent in biological papers?
line 759: What is meant by ‘metabolism of visual circulation’?
‘Circulation’ was replaced by ‘cycle’ (marked)
line 827: replace ‘analogical’ by ‘analogous’.
Word ‘analogical’ replaced by ‘analogous’

Reviewer 3 Report
The manuscript entitled « Impact of Microgravity and Other Spaceflight Factors on Retina of Vertebrates and Humans in vivo and in vitro » by Dr. Grigoryan is a timely and thoughtful review tying together the disparate studies on potentially deleterious effects of space travel on the visual system. My comments are posted below.
In a very general sense, the abbreviation for microgravity, µg, seems a bit confusing, since it is also the accepted abbreviation for micrograms. Isn’t it possible to use another expression, say µG ?
Also in a broad sense, although space flight obviously is mostly exposure to reduced gravitational pull, how does one take into account possible effects of the huge force exerted during take off ? I have seen studies where controls have remained on earth for the same duration, but these clearly lack this brief period of enormous g force from acceleration. For example, the study examing the effects of keeping medaka aboard the ISS do not evoke this point. The author does address this issue, and it seems to be very under researched.
And lastly in terms of general comments, the reader will have some difficulty in equating the detailed studies on retinal regeneration in newts, or development in birds, to what could be potential harmful effects for humans. Unless one is thinking of very long duration space missions in which female astronauts may become pregnant, it is hard to see how such experiments pertain to humans. I mean that the review should make more effort to discriminate between what are really basic research questions into how microgravity can influence developmental processes, and applications of limiting retinal stress in humans.
In the Introduction, 3rd para starting l. 42, the author mentions the few studies performed on animals but without giving any references at this point. This should be done, even if she returns to these in detail later in the review.
Considering effects of cosmic radiation, the author should take into account the surprising fact that many species show a rod photoreceptor-specific defect in DNA repair following exposure to gamma radiation (Frohns et al., 2014 Curr. Biol ; 2020 Cells). Defective repair seems to not occur in primates (presumably including humans), but could be mentioned as a problem in comparing data from animal models.
The following chapters are straight forward accounts of the sparse experiments investigating retinal changes in different vertebrate species, including humans. I guess luckily for us, the overall effects seem to be minor.
In conclusion, the review is a nice effort to tie together various studies on ocular consequences of space flight and radiation, though it doesn’t reveal any obvious general principles other than increased stress response and altered blood flow.
Overall the english is okay, but some improvement is necessary.
Author Response
Reviewer 3.
First and foremost, I express my gratitude and big appreciation to the reviewer for the work on the review paper and the comments made.
Changes and corrections have been made in compliance with the comments. All changes are indicated in the right margin of the paper. My answers to the comments are below in italic.
The manuscript entitled « Impact of Microgravity and Other Spaceflight Factors on Retina of Vertebrates and Humans in vivo and in vitro » by Dr. Grigoryan is a timely and thoughtful review tying together the disparate studies on potentially deleterious effects of space travel on the visual system. My comments are posted below.
In a very general sense, the abbreviation for microgravity, µg, seems a bit confusing, since it is also the accepted abbreviation for micrograms. Isn’t it possible to use another expression, say µG?
“Gravity” has 9 short forms. “g” and “G” – both are possible, however, in Wikipedia it’s written: “Micro-g environment (also μg, often referred to by the term microgravity)”. It is also the most frequent abbreviation present in corresponding papers. For this reason, I decided to keep the abbreviation μg.
And lastly in terms of general comments, the reader will have some difficulty in equating the detailed studies on retinal regeneration in newts, or development in birds, to what could be potential harmful effects for humans. Unless one is thinking of very long duration space missions in which female astronauts may become pregnant, it is hard to see how such experiments pertain to humans. I mean that the review should make more effort to discriminate between what are really basic research questions into how microgravity can influence developmental processes, and applications of limiting retinal stress in humans.
It is obvious that the previously conducted studies on fish, amphibians and birds were specifically biological, within the framework of the fundamental questions of gravitational biology, which seem now quite far from the direct problems of human stay in outer space. Direct transfer of the results to humans is impossible for many reasons and, first of all, because of evolutionarily fixed species features. However, the results are important in a broad biological context and as a basis for future research. I have included such a note in the text (section 4.2).
I also tried to reflect this in the chapter 4.2 and conclusion (lines 514-519;930-940).
In the Introduction, 3rd para starting l. 42, the author mentions the few studies performed on animals but without giving any references at this point. This should be done, even if she returns to these in detail later in the review.
It turned out to be extremely difficult for me to fulfill this recommendation of the reviewer, since it requires an extremely wide citation that does not directly relate to the issues discussed in my review. The sentence is a general point and well-known state of affairs in the field of gravitational biology and medicine. Further along the text I tried not to miss anything that I could find in a literature corresponding and/or directly relates to the essence of the review.
Considering effects of cosmic radiation, the author should take into account the surprising fact that many species show a rod photoreceptor-specific defect in DNA repair following exposure to gamma radiation (Frohns et al., 2014 Curr. Biol ; 2020 Cells). Defective repair seems to not occur in primates (presumably including humans), but could be mentioned as a problem in comparing data from animal models.
Thank you for this important remark. This is a fact additionally indicating different sensitivity of retinal cells to radiation exposure and then different ability to repair DNA, depending on the animal species, its age, radiation dose, etc. I have included additional fragment in the chapter 5 (marked).
The following chapters are straight forward accounts of the sparse experiments investigating retinal changes in different vertebrate species, including humans. I guess luckily for us, the overall effects seem to be minor.
Indeed, it seems so, and moreover, microgravity effect on the retina can be reversable/compensated. However, I was afraid to put this as a statement in the conclusions. The effect of microgravity depends on SF duration, individual characteristics, flight or ground-based experiment conditions, etc.
In conclusion, the review is a nice effort to tie together various studies on ocular consequences of space flight and radiation, though it doesn’t reveal any obvious general principles other than increased stress response and altered blood flow.
I am grateful for the comments, additions, and judgments. I would like to make one more remark. Rigorous and objective conclusions and general principles seem to require integrated and much more extensive knowledge of the issues discussed in the review. The paper gives the largely preliminary information that is available for today. More than half a century of research is apparently appeared insufficient, but presented studies are necessary and important basis for continuing research on SF-related changes in the visual system. This is what I tried to show in the review.

Round 2
Reviewer 1 Report
All of my suggestions are considered well.